# MODELING BOUNDEDLY RATIONAL AGENTS WITH LATENT INFERENCE BUDGETS

**Athul Paul Jacob**
MIT
apjacob@mit.edu

**Abhishek Gupta**
University of Washington
abhgupta@cs.washington.edu

**Jacob Andreas**
MIT
jda@mit.edu

## ABSTRACT

We study the problem of modeling a population of agents pursuing unknown goals subject to unknown computational constraints. In standard models of bounded rationality, sub-optimal decision-making is simulated by adding homoscedastic noise to optimal decisions rather than explicitly simulating constrained inference. In this work, we introduce a *latent inference budget model (L-IBM)* that models agents' computational constraints explicitly, via a latent variable (inferred jointly with a model of agents' goals) that controls the runtime of an iterative inference algorithm. L-IBMs make it possible to learn agent models using data from diverse populations of suboptimal actors. In three modeling tasks—inferring navigation goals from routes, inferring communicative intents from human utterances, and predicting next moves in human chess games—we show that L-IBMs match or outperform Boltzmann models of decision-making under uncertainty. Inferred inference budgets are themselves meaningful, efficient to compute, and correlated with measures of player skill, partner skill and task difficulty.

## 1 INTRODUCTION

Building effective models for multi-agent decision-making—whether cooperative or adversarial—requires understanding other agents' goals and plans. To help a friend navigate in a new environment, we must first understand where they want to go; to beat an opponent at chess, we must be able to predict their likely next moves. But decision-making, in humans and machines, is subject to computational constraints. Decision-makers often act suboptimally, relying on heuristics and approximations to choose their actions. Techniques that do not account for this suboptimality carefully may attribute behavior to differing intentions rather than different inference procedures.

How should we interact with agents seeking to accomplish unknown goals subject to unknown computational constraints? In standard models of bounded rationality (Luce, 1959), sub-optimal decision-making is simulated by adding noise to optimal decisions rather than explicitly simulating constrained inference. This results in models that treat agents as uniformly suboptimal in a way that fails to account for sub-optimal inference *algorithms* or for *non-homogenous* suboptimality.

In this paper, we describe a simple approach for building models of agents given traces of their behavior. Our approach explicitly models agents' "inference budgets", via a latent variable that controls the runtime of each agent's inference procedure. We show that for agents performing inference using *anytime algorithms* (algorithms that can be terminated at any point and return approximately correct solutions) inference budgets can be efficiently inferred from example behaviors. A diverse set of multi-agent decision-making procedures—including graph-based planning algorithms, recursive-rational models of human language production, and Monte Carlo tree search—admit imputation of inference budgets in this framework.

In three diverse agent modeling tasks—inferring navigation goals from routes, inferring communicative intents from human utterances, and predicting subsequent moves in human–human chess matches—we show that our approach matches or outperforms Boltzmann models of decision-making under uncertainty. Moreover, inferred inference budgets are themselves meaningful, correlating with measures of player skill, partner skill, and task difficulty. Our results show that sub-optimal human decision-making can be efficiently modeled with computationally constrained ver-

sions of standard search algorithms. By doing so, we obtain both accurate models of humans' decision-making and informative measures of their inferential capacity.

## 2 BACKGROUND AND PROBLEM FORMULATION

We study the problem of modeling one or more agents given given traces of their behavior. In particular, we assume that we observe a collection of trajectories (state–action sequences) produced by agents $\pi^* : s \mapsto a$ acting in a Markov decision process to maximize some reward function $R^*(\tau)$. Even when $R^*(\tau)$ is known to agents, inferring optimal actions is often intractable, so agents in the real world will in general *approximate* optimal behavior subject to some (unknown) computational constraints (which may differ from agent to agent). From this data, we seek to infer **agent models** $\pi$ defined in terms of (1) estimates $R$ of reward function $R^*$ (known to agents but not modelers), and (2) descriptions of the computational limitations that govern agents' choice of actions. In other words, we seek to model both *what agents wish to do* and *what agents will actually do* in any given state. Fig. 1 shows a conceptual example: assuming the agent receives a different reward for reaching each of the two goals, the three trajectories depicted there cannot be generated by the optimal policy for any reward function, but can be explained by model that can only look ahead to a limited number of positions in the maze.

Throughout this paper, we will model agent actions as arising from an **approximate inference procedure** $\pi(a \mid s; R, \beta)$ that takes as input a reward function and a **computational budget** $\beta$. We model agents by inferring values of $R$ and $\beta$ given the executed trajectories $\tau_i$.

The ability to infer goals from suboptimal (and even completely unsuccessful) plans is a key human skill, present in children as young as 18 months (Meltzoff, 1995). Computational models of bounded rationality thus have a long history in artificial intelligence, cognitive science, and behavioral economics. But what does this suboptimality look like in practice, and how should we model and infer the inference budget $\beta$ simply from observations of behavior?

One of the most widely used models of boundedly rational decision-making is the so-called **Boltzmann** model (Luce, 1959), in which agents take actions according to

$$\pi(a \mid s; R, \beta) \propto \exp\{\beta \cdot R(s, a)\} \quad (1)$$

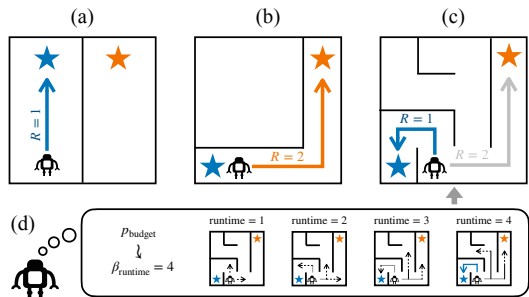

Figure 1: Inferring rewards from boundedly-rational trajectories. The agent will move to the blue star (a), but prefers to move toward the orange star when both are available (b). When locating the orange star requires solving a harder search problem, however, the agent seeks the blue star instead, indicating that its search abilities are limited (c). Our proposed approach automatically infers the budget that the agent uses when planning (d). Knowing this budget, we could perhaps assist this agent by providing a targeted hint (*move right*) at the beginning of its trajectory.

This equation has a number of appealing interpretations, e.g. as the policy that achieves a target reward while maximizing entropy. It has been used to model not just the selection of actions, but also trajectories, preferences, corrections, and more—see Jeon et al. (2020) for a recent survey. More elaborate approaches in this family also predict $\beta$ conditioned on the current state or action history, making it possible to model state-dependent skill (Beliaev et al., 2022).

However, Boltzmann models have a significant limitation: the probability of generating an action in Eq. (1) depends only on the true value of that action, and not on the cost of acquiring a high-quality value estimate in the first place. To see why this might be a problem, consider again the trajectories depicted in the conceptual example Fig. 1(b–c), which differ *only* in the difficulty of the search problem, and not in the cost of the optimal trajectory at all. A model of boundedly rational decision-making with the form of Eq. (1) cannot account for this difference.

There is a large body of other approaches on modeling human planning under resource constraints in psychology, economics and in classical AI (Callaway et al., 2022; Russell & Wefald, 1991; Huys et al., 2015; 2012; Camerer et al., 2004; Griffiths et al., 2019; Boddy & Dean, 1989; van Opheusden et al., 2023, inter alia). Notably, Zhi-Xuan et al. (2020) build an explicit hierarchical Bayesian model of a symbolic planning procedure while inferring per-timestep inference budgets.

In general, these approaches make strong assumptions about how planning is performed. Here, we seek to develop a general framework that avoids strong assumptions about either the functional form of the reward model or the algorithmic form of the planning procedure. Related approaches were proposed by Evans & Goodman (2015) and Shah et al. (2019), though with simpler environments or inference procedures. The framework we develop can be applied to model real-world behavior in tasks as diverse as language generation and chess gameplay.

## 3 Inferring Rewards and Inference Budgets from Behavior

As motivated in Section 2, our goal is to model agents acting to optimize an unknown value function subject to an unknown computational constraint. In practice, we often want to model populations comprising multiple agents or agent sub-populations $(\pi_1^*, \pi_2^*, \ldots \pi_N^*)$ with a shared reward function $R^*$ (e.g. winning at chess) but differing computational constraints.

To do so, we assume we have access to a collection of trajectories $\{\tau\}_i = \{\tau_i^1, \tau_i^2, \ldots \tau_i^{M_i}\}$, with each collection of trajectories $\{\tau\}_i$ generated by a different agent or sub-population $i$. We model these trajectories as drawn from the following generative process:

1. at each timestep, agent $i$ draws a budget $\beta$ from an agent-specific prior $p_{\text{budget}}(\beta \mid \eta_i)$
2. $\pi_i^*$ chooses actions according to a budget-constrained inference procedure $\pi_i^*(a \mid s; R^*, \beta)$

Because budgets may vary between trajectories, learning a model of these agents ultimately consists of learning reward parameters $\theta$ and agent-specific budget-generating parameters $\eta_i$ while *marginalizing* over latent budgets themselves. We do so via maximum *a posteriori* inference, optimizing:

$$\arg\max_{\theta, \eta} \sum_{\substack{i \\ \tau \in \{\tau\}_i \\ (s,a) \in \tau}} \log \pi(a \mid s; \theta, \eta) = \arg\max_{\theta, \eta} \sum_{\substack{i \\ \tau \in \{\tau\}_i \\ (s,a) \in \tau}} \log \sum_{\beta} p_{\text{budget}}(\beta \mid \eta_i) \cdot \pi(a \mid s; R_\theta, \beta) \quad (2)$$

If $\pi(a \mid s; R^*, \beta)$ is an arbitrary inference algorithm, Eq. (2) might present a challenge: this inference procedure must be run for all possible values of $\beta$, which will in general be intractable. Under what circumstances can we optimize this equation efficiently? The key observation in this paper is that if $\pi$ is an *anytime inference algorithm* (Dean & Boddy, 1988), we can evaluate $n$ values of $\beta$ as quickly as we can evaluate one, making this optimization tractable.

**Definition 1.** *An anytime algorithm $\pi$ is one that runs for $t$ timesteps and produces a sequence of inference states $(f_1, f_2, \ldots f_t)$, where every $f_i$ can be computed from $f_{i-1}$ in $\mathcal{O}(1)$ time, and $f_i$ can be used to select an action according to some $\pi(a \mid s; R, f_i)$.*

As we will see shortly, many canonical inference algorithms used in single- and multi-agent decision-making scenarios have this from. In these cases, rather than letting the budget parameter $\beta$ determine noise or suboptimality, we may use it to parameterize the runtime of the agent's inference procedure itself, writing:

$$\log \pi(a \mid s; \theta, \eta_i) = \log \sum_{\beta} p_{\text{budget}}(\beta_{\text{runtime}} \mid \eta_i) \cdot \pi(a \mid s; R_\theta, f_{\beta_{\text{runtime}}}) \quad (3)$$

where we have denoted the budget $\beta_{\text{runtime}}$ to indicate that it parameterizes the runtime of the anytime inference algorithm. Crucially—by definition—computing this sum up to some maximum $\beta$ requires no more time than computing its final term.

The remainder of this paper looks at instantiations of this basic modeling framework in three different domains. In Section 4, we study the problem of inferring navigation goals from maze domain using a truncated graph search algorithm. In Section 5, we study rational speech acts (RSA) for inferring communicative intents from human utterances. Finally, in Section 6, we model human action prediction in chess using Monte-Carlo tree search (MCTS).

## 4 Solving Mazes with Truncated Depth-First Search

We begin with a pedagogical example of latent inference budget model applied to a simple, single-agent decision-making task: maze navigation. Agents are placed at a random position in a maze with

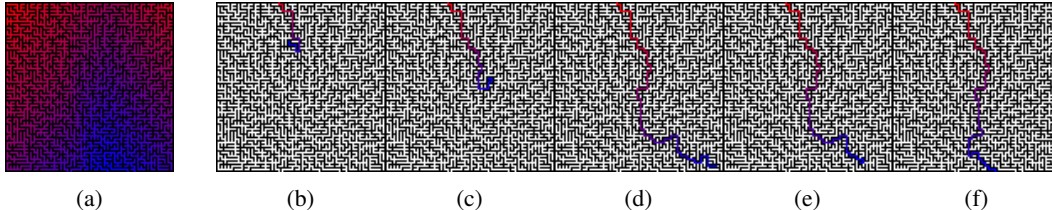

|     |     |     |     |     |     |
| --- | --- | --- | --- | --- | --- |
| (a) | (b) | (c) | (d) | (e) | (f) |

Figure 2: Examples of the maze task. (a) Example of the value function heuristic applied to each state in the maze. Red indicates low value states and blue indicates high value states. (b)-(f) depicts the example trajectories of agents with depth budgets of 1, 2, 5, 10 and 20.

five exits. Each exit is a state $e_i$ associated with a reward $R_i$. Agents attempt to navigate toward the highest scoring exit by taking navigation actions (north, east, south, west). Here our goal is to recover the rewards $R_i$ that a single agent associates with each exit, along with agent budget parameters $\eta$, given observations of the agent's behavior.

## 4.1  AGENT MODEL

We assume that agents select navigation actions using a heuristic with a known functional form, in which the value of a state $s$ is approximated as:

$$V(s) = \frac{\sum_i R_i e^{-\|s-e_i\|_1 \cdot R_i}}{\sum_i e^{-\|s-e_i\|_1 \cdot R_i}} \tag{4}$$

where $\|s - s'\|_1$ measures the Manhattan distance between a pair of states (i.e. maze positions). Intuitively, we model agents as "attending" to each exit in proportion to both its distance and associated reward. We assume that agents use this heuristic to perform **truncated breadth-first search**. In a state $s$, agents first estimate the value of each action $a$ by computing the value of the best state reachable in $\beta_{\text{runtime}}$ actions, starting with $a$. Formally:

$$Q_{\text{runtime}}(a \mid s) = \max_{\tau:\tau_0=a, |\tau|=\beta_{\text{runtime}}} V(\tau_{\beta_{\text{runtime}}}) \tag{5}$$

where $\tau_0$ and $\tau_{\beta_{\text{runtime}}}$ respectively denote the first and last actions in the trajectory $\tau$. Finally, agents select actions in proportion to these Q-values (Haarnoja et al., 2017):

$$\pi(a \mid s; \beta_{\text{runtime}}, R) \propto e^{Q(a|s)} \tag{6}$$

With this agent parameterization, Eq. (2) can be computed efficiently:

**Proposition 1.** *Truncated breadth-first Search (TBFS) is an anytime inference algorithm. (Represent each inference state $f_\beta$ as the set of frontier states and values reachable from each starting action. To compute $f_{\beta+1}$, add the unexplored children of these states to the set.)*

## 4.2  DATA

In this pedagogical example, we treat the agent model in Section 4.1 as the true data-generating process. We fix a set of parameters $R_i$ and $\beta_{\text{runtime}}$, generate a collection of synthetic trajectories using Eq. (6), then attempt to recover these parameters using Eq. (2). (This allows us to validate the feasibility of our approach under ideal conditions—later sections will apply it to real datasets of human-generated actions). In particluar, we generate 5 agents with runtime budgets of 1, 2, 5, 10, and 20 respectively. Example trajectories from each of these agents are depicted in Fig. 2.

## 4.3  EVALUATION

We compare L-IBMs with a Boltzmann model in which agents select actions according to:

$$Q_{\text{temp}}(a \mid s) = \beta_{\text{temp}} \cdot \max_{\tau:\tau_0=a} R(\tau) \tag{7}$$

where $R(\tau)$ denotes the *final* reward obtained along the complete trajectory $\tau$ (i.e. upon reaching some exit $R_i$). We also compare to simple baselines in which the agent performs truncated search up to a constant (not inferred) depth. We evaluate these models in two ways:

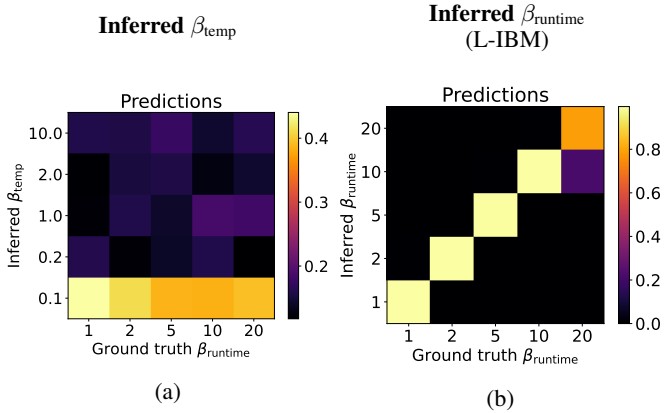

|            | (a)        |            | (b)        |
|------------|------------|------------|------------|

Figure 3: Inferred parameters $\eta_i$ (distributions over $\beta$) for the maze navigation task. (a) L-IBM almost perfectly recovers these parameters, while (b) the Boltzmann model shows no significant differences across inferred $\beta_{\text{temp}}$.

| Approach | Accuracy |
|----------|----------|
| $\beta_{\text{runtime}} = 0$ | 5 |
| $\beta_{\text{runtime}} = 20$ | 16 |
| Inferred $\beta_{\text{temp}}$ | 20 |
| L-IBM | **44** |

Table 1: Agent action prediction accuracies in maze navigation. L-IBM significantly outperforms baselines.

**Predicting actions.** In held-out states, we evaluate models' **exact-match** accuracy in predicting an agent's next action. Results are shown in Table 1. Models that assume a constant depth perform worst. While Boltzmann models are better able to predict agents' next actions than these fixed-budget models, they are significantly outperformed by L-IBM.

**Predicting rewards.** We also evaluate whether inferred prior distributions over $\beta$ recover the true values used to generate the data. Results for L-IBM and the Boltzmann model are shown in Fig. 3a. It can be seen that L-IBM almost perfectly recovers these parameters (suggesting that prediction errors in Table 1 result entirely from errors in the inferred reward parameters $R_i$). Meanwhile, the Boltzmann model shows no significant differences in inferred $\beta_{\text{temp}}$ across depth budgets, emphasizing the discrepancy between the two mdoels of suboptimality.

Together, these results show that L-IBM is computationally tractable and capable of making accurate predictions and inferring meaningful parameters in simple search problems. In the remainder of the paper, we apply it to modeling real human behavior in more complex decision-making tasks.

## 5 PRAGMATIC LANGUAGE UNDERSTANDING WITH RATIONAL SPEECH ACTS

The next task we consider focuses on **pragmatic language understanding**—inferring speakers' communicative intents from their utterances. Humans readily produce and understand language in ways that deviates from its "literal" meaning. In Table 2, for example, a color that would be described on its own by most speakers *purple* is instead labeled *blue* in some contexts. A large body of work in cognitive science models this kind of context-based language understanding as the result of an iterative inference process (Frank & Goodman, 2012; Franke, 2013): for example, in Row 2 of Table 2, a speaker might choose to describe the highlighted color as *blue* by reasoning that a naïve listener might resolve *purple* to the second color in the row. A more sophisticated listener, in turn, can predict this speaker behavior, and successfully infer the intended meaning. But this kind of recursive reasoning about other agents can be computationally demanding, and requires sophisticated internal models of other language users. Experimental evidence suggests that is deployed selectively, and to different degrees by different language users (Franke & Degen, 2016). We use L-IBMs to determine when, and to what extent, this kind of recursive reasoning is used during language production.

| Context | | | Utterance |
|---------|---|---|-----------|
| 1. | | | *purple* |
| 2. | | | *blue* |
| 3. | | | *blue* |

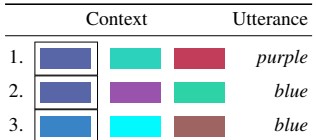

Table 2: Example of the reference color (within the black box) and the two distractor colors, along with the utterance produced by a speaker from the colors in context task (Monroe et al., 2017). Notice how the context affects the utterance, even as the reference color remains fixed.

Our experiments focus on a **reference game** of the kind depicted in Table 2 (Monroe et al., 2017). Reference games are a staple of research on pragmatic language use. In a reference game, both a listener and speaker observe a set of candidate referents (e.g. colors). The speaker is privately given one of the colors as a target; they must then produce a natural language utterance for the listener. Finally, the listener selects a color patch, and both players win if they agreed on the target.

By fitting an L-IBM to utterances and choices in human reference games, we investigate (1) whether we can infer whether humans are engaged in pragmatic reasoning from behavior alone, (2) whether there are differences between players in their ability to reason about their interlocutors, and (3) whether these differences actually predict communicative success (i.e. whether players with greater inference budgets are better at making themselves understood).

## 5.1 AGENT MODEL

We build on the Rational Speech Acts (RSA) model of Frank & Goodman (2012). This model frames communication as one in which Bayesian listeners and speakers reason recursively about each others' beliefs in order to select utterances and actions. The starting point of RSA is a **literal listener** $\pi_L^0$ that maps utterances $u$ to actions according to their non-contextual meanings. (In Table 2, a literal listener hearing the word *purple* might choose randomly between the first two colors in the second row, as both would be reasonably described as purple out of context.) The literal listener may be implemented by any model (e.g. a lookup table or a neural network; Andreas & Klein, 2016) with parameters $\theta$. Next, given a reference target $t$, a **pragmatic speaker** $\pi_S$ chooses an utterance in proportion to the probability that it will cause a literal listener to take the right action:

$$\pi_S^1(u \mid t) \propto p(\pi_L^0 \text{ selects } t \text{ upon hearing } u) = \pi_L^0(t \mid u) \tag{8}$$

(RSA speakers are standardly parameterized with an additional Boltzmann rationality parameter, which we will discuss momentarily.) Finally, **pragmatic listeners** observe speaker utterances $u$, and reason about which reference targets were most likely to have produced those utterances:

$$\pi_L^1(t \mid u) = p(\pi_S^1 \text{ intends to signal } t \mid u) \propto \pi_S^1(u \mid t)\, p(t) \tag{9}$$

Crucially, this process may be repeated, with speakers $\pi_S^i$ reasoning about ever-more-sophisticated speakers $\pi_L^{i-1}$, etc. But how many rounds of iteration actually explain human behavior? In the latent inference budget model framework, we may model this by embedding RSA inside an L-IBM, with the budget $\beta$ parameterizing the number RSA iterations performed by each agent:

$$\pi_S(u \mid t; \theta, \eta) = \sum_\beta \beta_{\text{runtime}}(\beta \mid \eta)\pi_S(u \mid t; \theta, \beta) \tag{10}$$

$$\pi_S(u \mid t; \theta, \beta) = \pi_S^\beta(u \mid t) \tag{11}$$

(and analogously for $\pi_L$.)

**Proposition 2.** *Rational Speech Acts (RSA) is an anytime inference algorithm. (Each inference state $f_\beta$ is $\pi_S^\beta$ or $\pi_L^\beta$. Each of these can be derived from the other in constant time via Eqs.8–9.)*

## 5.2 DATA

For this task, we use the data collected by Monroe et al. (2017). Each game consists of roughly 50 rounds played between a human speaker and a human listener. In each round, the speaker observes a target color along with two distractors. The speaker produces an utterance and the listener has to click on one of the colors. The dataset consists of 46,994 rounds across 948 games. We create a 80/10/10 split across train, valid and test sets. Monroe et al. stratify the dataset into three difficulties (easy, medium and difficult) based on perceptual similarity between colors and distractors. Because each game is annotated with a unique identifier for both the speaker and the listener, we may further stratify the dataset according to *player skill*: we compute the fraction of games won by each (speaker, listener) pair, then group these pairs into six buckets according to their win rate percentile relative to other players. This allows us to examine the relationship between inference budget and both task difficulty and communicative success.

## 5.3 MODELS

Following Monroe et al. (2017), we implement the literal listener $\pi_L^0$ using a transformer model that receives all three colors (represented as HSL vectors) and a natural language utterance as input, and predicts the index of the target color as output. We embed this listener model within the speaker–listener recursion defined by Eq. (9), then train it end-to-end (with budget parameters $\eta_i$) on the Colors in Context data using Eq. (2).

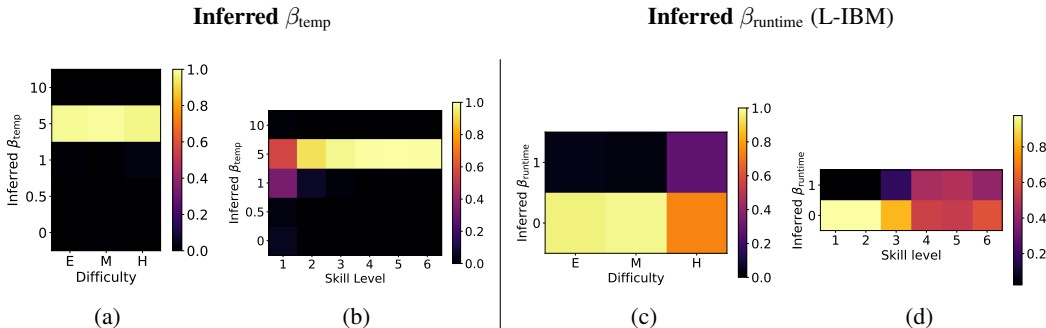

Figure 5: Inferred distributions over $\beta$ in RSA. X-axis indicates the difficulty level (**E**asy, **M**edium, **H**ard) or the player skill level (between 1 and 6, 6 being the most skilled players). The inferred $\beta_{\text{temp}}$ across difficulty in a) and player skill in b) is not as meaningful as it is for $\beta_{\text{runtime}}$ in c) and d). c) When separating games by difficulty, L-IBM infers that the non-literal speaker is employed only for the hardest condition. d) When separating games by player skill, we infer that the weakest players can be modeled exclusively as literal speakers, while stronger players can be modeled as a mix of literal and pragmatic speakers.

The constant of proportionality in Eq. (8) involves a sum over all natural language strings, which is cannot be computed efficiently. Here, also following (Monroe et al., 2017), we perform a sampling-based approximation: we train a *literal speaker* model to generate plausible utterances, then sum over a finite number of such samples to obtain a distribution over strings. See McDowell & Goodman (2019) for more details. The literal speaker is parameterized identically to the literal listener, but outputs strings rather than color indices.

In experiments investigating the relationship between task difficulty and inference budget, we fit one $\eta_i$ *per condition* (easy, medium, hard). In experiments investigating the relationship between communicative success and inference budget, we fit one $\eta_i$ *per skill level* (between 1 and 6).

### 5.4 EVALUATION

Standard implementations of RSA modifies Eq. (8) to include a Boltzmann parameter for speakers:

$$\pi_S^i(u \mid t; \beta) \propto \exp\{\beta_{\text{temp}} \log \pi_L^{i-1}(t \mid u)\} \tag{12}$$

Like our $\beta_{\text{runtime}}$, this parameter is intended to model possibly sub-optimal behavior on the part of speakers and listeners. We compare an L-IBM to a model of this form. In particular, we fix the number of RSA iterations to one, use the same data as above to estimate literal listener parameters jointly with a prior distribution over $\beta_{\text{temp}}$:

$$\pi_S^1(u \mid t; \theta, \eta) = \sum_\beta p_{\text{temp}}(\beta \mid \eta)\pi_S^1(u \mid t; \beta) \tag{13}$$

where $\pi_S^1$ is defined as in Eq. (12).

Table 3 shows different models' ability to predict the target referent given human speaker utterances. Consistent with the findings of (Monroe et al., 2017), because even literal models have access to all three referents, all model variants can achieve good task performance. When we look at inferred values for $\beta_{\text{runtime}}$ and $\beta_{\text{temp}}$, however, we begin to see significant differences between models. When stratifying games by *difficulty*, we infer that the non-literal speaker is employed only for the hardest conditions. When stratifying games by *player skill*, we infer that the weakest players can be modeled exclusively as literal speakers, while stronger players can be modeled as a mix of literal and pragmatic speakers. To the best of our knowledge, this is the first example of an RSA-type model being used to infer individual differences in pragmatic language use within a speaker population; we expect that these tools may be of independent interest to the cognitive science community. Additional experiments, predicting the object that the listener picked instead of the one the speaker is presented can be found in Appendix C.

## 6 PLAYING CHESS WITH MONTE-CARLO TREE SEARCH

Finally, we turn from cooperative to adversarial decision-making tasks. We focus on chess, a popular two-player sequential game widely used as a benchmark for AI systems. Here, we are interested in

| Model | Type | Accuracy |
|---|---|---|
| $\beta_{\text{runtime}} = 0$ | - | 83.3 |
| $\beta_{\text{runtime}} = 1$ | - | 83.0 |
| Inferred $\beta_{\text{temp}}$ | player skill | 83.9 |
| Inferred $\beta_{\text{runtime}}$ (L-IBM) | player skill | **84.0** |
| Inferred $\beta_{\text{temp}}$ | difficulty | 83.5 |
| Inferred $\beta_{\text{runtime}}$ (L-IBM) | difficulty | 82.7 |

Table 3: Performance of different RSA models in predicting the speaker target. All models (including literal models and fixed-depth RSA models) achieve similar predictive performance—because even literal models have access to all three referents, all model variants can achieve good task performance. $\beta_{\text{runtime}} = 0$ represents the base literal listener.

| Model | Type | Accuracy |
|---|---|---|
| IL | - | 42.06 |
| $\beta_{\text{runtime}} = 100$ | - | 43.64 |
| Inferred $\beta_{\text{puct}}$ | Active Elo | 43.77 |
| Inferred $\beta_{\text{runtime}}$ (L-IBM) | Active Elo | **44.17** |
| Inferred $\beta_{\text{puct}}$ | Opponent Elo | 43.84 |
| Inferred $\beta_{\text{runtime}}$ (L-IBM) | Opponent Elo | **44.17** |
| Inferred $\beta_{\text{puct}}$ | Time Control | 43.61 |
| Inferred $\beta_{\text{runtime}}$(L-IBM) | Time Control | **44.15** |

Table 4: Accuracy of predicting an agent's next action in chess. Models with MCTS outperform the depth-0 (imitation learning) baseline. Learning subpopulation-specific $\beta$ enhances performance, with L-IBM-based learning of $\beta_{\text{runtime}}$ consistently outperforming $\beta_{\text{puct}}$ by a slight margin.

modeling human chess play—specifically, observing data from a population of sub-optimal agents with a common reward function (winning the game) and attempting to infer those agents' computational constraints. In human human play, there can be numerous sources of such constraints: a player paired against a strong opponent will likely to plan for longer than against a weaker opponent; some variants (like blitz chess) deliberately limit players' time-per-move (and, we might expect, the quality of their plans). Given a dataset of human games played under different time constraints and player strengths, can we use L-IBM to model variability in players' decisions across game states?

## 6.1 AGENT MODEL

In this work, we model chess players as selecting actions using **Monte Carlo tree search** (MCTS). Recent work (Jacob et al., 2022) has shown that MCTS is a good model of strong human players. Here, following (Silver et al., 2018; 2016; Jacob et al., 2022; Grill et al., 2020), we implement one of the most common modern forms of MCTS, which uses a value function $V$ predicting the expected total future reward and a policy prior $\pi^0$ to guide exploration. At a high level, MCTS operates by incrementally growing a game tree starting at the root node, repeatedly picking some path to explore down the tree, performing a value function evaluation and then walking back up the tree updating all the value estimates based on that result. At each node, MCTS treats action selection as a multi-armed bandit problem. We use a standard exploration policy (Kocsis & Szepesvári, 2006): during inference at each node of the search tree, we choose actions according to:

$$\arg\max_a Q_t(a \mid s) + \beta_{\text{puct}}\pi^0(a \mid s)\left(\sqrt{\textstyle\sum_b N(s,b)}\right)/\left(N(s,a) + 1\right) \qquad (14)$$

where $Q_t(s, a)$ is the estimated expected future reward for $i$ from playing action $a$ in state $s$ at iteration $t$, the visit count $N(s, a)$ is the number of times $a$ has been explored from $s$, $\pi^0(a \mid s)$ is an "anchor" policy, and $\beta_{\text{puct}}$ is a tunable parameter trading off exploration versus exploitation. After expanding $\beta_{\text{runtime}}$ nodes of this tree, an agent's final action is sampled from a distribution:

$$\pi(a \mid s; \beta_{\text{runtime}}) = \beta_{\text{puct}} \frac{\sqrt{\beta_{\text{runtime}}}}{N(s,a) + 1} \frac{\pi^0(a|s)}{\gamma - Q_{\beta_{\text{runtime}}}(a \mid s)} \qquad (15)$$

where $\gamma$ is chosen such that $\pi$ forms a proper probability distribution.

**Proposition 3.** *MCTS is an anytime inference algorithm. (Let each inference state $f_\beta$ be the tree of nodes and visitation counts after $\beta$ evaluations. This tree is refined by evaluating Eq. (15) once.)*

With $\pi(a \mid s; \beta_{\text{runtime}})$ as defined above, we may instantiate an L-IBM for MCTS:

$$\pi^{\text{runtime}}(t|u; \eta, \theta) = \sum_{\beta_{\text{runtime}}} p_{\text{budget}}(\beta_{\text{runtime}} \mid \eta_i) \cdot \pi(a; s, \beta_{\text{runtime}}) \qquad (16)$$

We train the base initial policy $\pi_0$ and a value model $\tilde{v}_0$ as two different output heads of a deep neural network using imitation learning. Our architecture is a 4-block residual network similar to those used in prior work (McIlroy-Young et al., 2020). Unlike previous sections, we do not learn the value functions jointly with $p_{\text{budget}}$. Instead, we first learn a single value function, then fit $p_{\text{budget}}(\beta_{\text{puct}} \mid \eta_i)$ and $p_{\text{budget}}(\beta_{\text{runtime}} \mid \eta_i)$. We stratify players into sub-populations according to player Elo (a proxy for player skill), and opponent Elo and time control (both proxies for task difficulty). As in Section 5, we estimate a separate $\eta_i$ for each group within each stratified dataset.

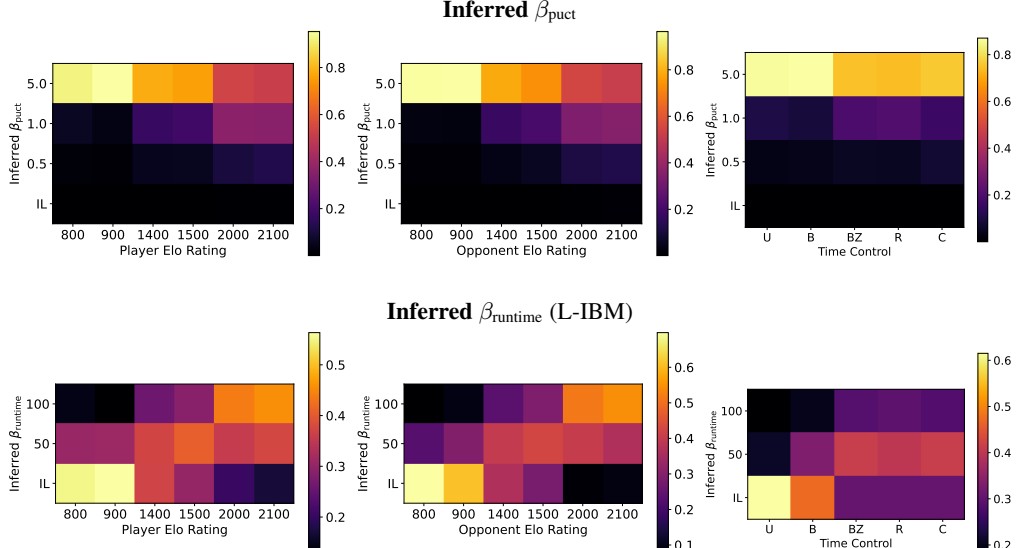

Figure 6: Inferred distributions over $\beta$ in Chess using MCTS. The horizontal axis indicates the player Elo rating, opponent Elo rating buckets and time control: Ultra Bullet (U), Bullet (B), Blitz (BZ), Rapid (R) and Classical (C). The top row depicts the distributions for $\beta_{\text{puct}}$ and the bottom row depicts the distributions for $\beta_{\text{runtime}}$. When the player's strength, opponent's strength, or time increases, $\beta_{\text{runtime}}$ infers greater runtime.

## 6.2 DATA

We use similar data to previous models of human chess play (McIlroy-Young et al., 2020): First, a dataset $D_{\text{large}}$ containing roughly 6 million moves, used to train the base value function; second, a dataset $D_{\text{small}}$ containing roughly 75,000 moves, used to build population-specific models. $D_{\text{small}}$ includes metadata describing players' Elo ratings (a measure of strength) and game formats (the amount of time players had to select moves). See Appendix B for details.

## 6.3 EVALUATION

Unlike in the two domains studied above, there is already an established literature on modeling sub-optimal behavior via MCTS outside the Boltzmann framework. The most successful current approach models individual differences in play (Jacob et al., 2022) by fitting $\beta_{\text{puct}}$. We thus compare to a baseline in which $\eta_i$ parameterizes a distribution over values of $\beta_{\text{puct}}$ rather than tree expansions.

Accuracy (in terms of top-one predictions and negative log-likelihood) is reported in Table 4. As in past work, we find that models that with explicit search outperform imitation-learning baseline. Learning sub-population specific $\beta$ improves the performance even further, with L-IBM-based learning of $\beta_{\text{runtime}}$ consistently outperforming $\beta_{\text{puct}}$ by a small margin.

Inferred budget parameters are shown in Fig. 6. Here, we observe that as the player strength or the opponent strength increases as measured by the Elo ratings, $\beta_{\text{runtime}}$ infers higher runtime. We also observe the same as the time control increases: $\beta_{\text{runtime}}$ infers higher runtime as the duration of each move of the game increases. $\beta_{\text{puct}}$ shows a weaker, but similar trend: as the agents or opponents get stronger, or as the time control increases, $\beta_{\text{puct}}$ infers lower values of $\beta_{\text{puct}}$, indicating that players are deviating from the prior and are relying more on the search Q-values.

## 7 CONCLUSION

We have described latent inference budget models, a family of approaches for modeling agents acting to achieve unknown goals subject to unknown constraints on their inferential capabilities. Instead of assuming either global optimality of decision-making or uniform suboptimality, our approach explicitly infers the runtime that agents devote to approximate inference. This paradigm is applicable to all anytime inference algorithms. In three domains—maze navigation, pragmatic language understanding, and playing chess—we demonstrated that it can outperform classical models of bounded rationality while imputing meaningful measures of human skill and task difficulty.

## ACKNOWLEDGEMENTS

This work was supported by the National Science Foundation under grants IIS-2238240, IIS-2212310, and a seed grant from the MIT Schwartzman College of Computing "Artificial Intelligence for Augmentation and Productivity" program. Thanks to Jennifer Hu for helpful discussions about modeling individual differences in Rational Speech Act models.

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

# A    TRAINING HYPERPARAMETERS

We will detail the training hyperparameter details in this section.

## A.1    MAZE

All models in Section 4 were trained using the Adam optimizer (Kingma & Ba, 2015), where the learning rates were swept across the following values $[1.0, 0.5, 1e-1, 0.05, 1e-2, 5e-3, 1e-3, 5e-4, 1e-4, 5e-5]$ for 50 epochs. The values presented in Table 1 were picked from the model with the best validation accuracy across the learning rates. We implemented truncated BFS using Pytorch (Paszke et al., 2019) and Numpy (Harris et al., 2020) and we also used the mazelib library to generate the data.

## A.2    COLORS IN CONTEXT

The models trained in Section 5 are based on the transformer architecture and trained from scratch. The speaker model was trained based on the T5 model (Raffel et al., 2020) with the following hyperparameters described in Table 5. The speaker was trained with a batch size of 64 using the Adam optimizer with learning rate $1e-4$ for 25 epochs.

| Parameter | Value |
|---|---|
| Number of Layers | 4 |
| Number of Heads | 4 |
| Model Dimension | 32 |
| Key-Value Dimension | 16 |
| Feedforward Dimension | 32 |

Table 5: Hyperparameter configuration of the speaker model based on T5 (Raffel et al., 2020).

| Parameter | Value |
|---|---|
| Hidden Size | 64 |
| Number of Hidden Layers | 4 |
| Number of Attention Heads | 4 |
| Intermediate Hidden Size | 256 |

Table 6: Hyperparameter configuration of the listener model based on BERT.

All the listener models were based on the BERT (Devlin et al., 2019) model with the configuration described in Table 6. The listener models were trained using Adam and the learning rates were swept across the following values $[1e-3, 5e-4, 1e-4, 5e-5]$ for upto 50 epochs. The values presented in Table 8 and Table 3 were picked from the model with the best validation accuracy across the learning rates. We trained the models using Pytorch (Paszke et al., 2019) and Huggingface (Wolf et al., 2020) libraries. We specifically implemented RSA using Pytorch.

## A.3    CHESS

The value and policy network used in Section 6 are based on an architecture that is a 4-block residual network similar to those used in prior work McIlroy-Young et al. (2020); Jacob et al. (2022); McIlroy-Young et al. (2022). The policy and value network was trained using Adam with a learning rate of 0.001, a batch size of 4096 and for upto 30 epochs. The epoch used in the rest of the section was picked based on the validation accuracy.

In the second set of fine-tuning experiments, for every set of conditioning type, a simple feedforward network was trained using Adam with a batch size of 512. The models in Section 6 were picked by selecting the learning rates between $1e-3, 5e-4, 1e-4, 5e-5$ with the best validation accuracy.

For chess, the base policy and value functions were trained using Ray library (Liang et al., 2017) and Pytorch. MCTS was specifically implemented using Numpy. We also used the pettingzoo library for simulating moves.

# B    CHESS DATA

$D_{\text{large}}$ consists of 5,974,872 moves in the training split, 60,968 in the validation split and 60,969 moves in the test set. These data points were randomly sampled from the January, 2019 database

release of a chess website (lichess). $D_{small}$ consists of 50,000 moves in the training split, 12,041 moves in the validation split and 12,040 moves in the test split. These data points were randomly sampled from the February, 2019 lichess database release but filtering such that only those players with Elo ratings in the following buckets were considered: [800-1000], [1400-1600] and [2000-2200].

The dataset contains 5 different types of time control. In increasing duration, they are **Ultra Bullet**, **Bullet**, **Blitz**, **Rapid** and **Classical** (see Table 7).

| Time control | Estimated Duration (seconds) |
|---|---|
| UltraBullet | $< 29$ |
| Bullet | $< 179$ |
| Blitz | $< 479$ |
| Rapid | $< 1499$ |
| Classical | $\geq 1500$ |

Table 7: Estimated game durations across different time controls.

The time controls used in our work have estimated durations that are defined in Table 7:

## C    ADDITIONAL EXPERIMENTS: COLORS IN CONTEXT

In this section, we include additional experiments for the pragmatics domain where we train the models to predict the object that the listener picks. We present the results of a similar set of experiments as in Section 5 in Table 8 and Fig. 8. We specifically note that the inference based approaches outperform the baselines in this setting.

| Model | Type | Accuracy |
|---|---|---|
| $\beta_{runtime} = 0$ (Literal listener) | - | 80.4 |
| $\beta_{runtime} = 1$ | - | 81.8 |
| Inferred $\beta_{temp}$ | player skill | 82.3 |
| Inferred $\beta_{runtime}$ (L-IBM) | player skill | **83.1** |
| Inferred $\beta_{temp}$ | difficulty | 82.7 |
| Inferred $\beta_{runtime}$ (L-IBM) | difficulty | 82.1 |

Table 8: Performance of different RSA models in predicting the speaker target. The $\beta$ based models outperform the baseline models: literal models and fixed-depth RSA models.

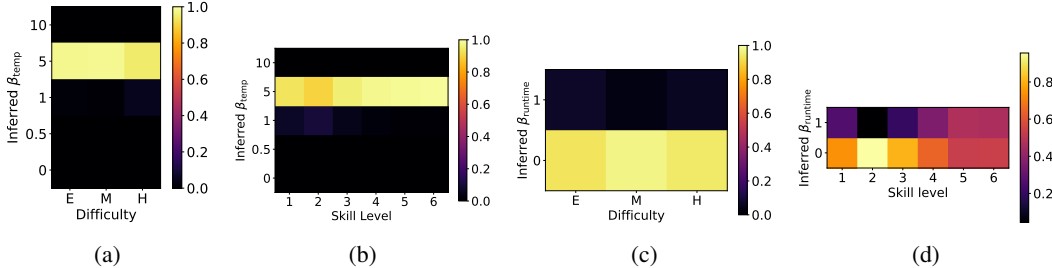

    (a)              (b)             (c)            (d)

Figure 8: Inferred distributions over $\beta$ in RSA, with the listener target. X-axis indicates the difficulty level (**E**asy, **M**edium, **H**ard) or the player skill level (1 - 6, 6 being the most skilled players). The inferred $\beta_{temp}$ across difficulty in a) and player skill in b) is not as meaningful as it is for $\beta_{runtime}$ in d). When separating games by player skill, we infer that the weakest players can be modelled with a smaller mix towards pragmatic speakers compared to stronger players

# D  ADDITIONAL DISCUSSION

## D.1  RELATIONSHIP BETWEEN $\beta_{\mathrm{puct}}$ AND ELO RATING IN CHESS

$\beta_{\mathrm{puct}}$ (as used in popular strength-modeling approaches like Silver et al. (2016)) doesn't simply control exploration vs. exploitation. Instead, it biases exploration towards an initial policy prior $\pi^0$. As $\beta_{\mathrm{puct}}$ tends to infinity, it is identical to playing $\pi^0$. When $\beta_{\mathrm{puct}}$ is set to 0, it is equivalent to greedily picking the search Q-values. In Fig. 6 and as it relates to ELO rating, we notice that stronger players start deviating more from their base policy $\pi^0$ to instead depend more on their MCTS search Q-values. Therefore indicating that stronger players rely more on search compared to weaker players.

