# OpenReview forum: "Modeling Boundedly Rational Agents with Latent Inference Budgets"
_ICLR.cc/2024/Conference — ICLR 2024 poster_

### Official Review · Reviewer_q6an · 2023-10-29

**Soundness:** 2 fair
**Presentation:** 2 fair
**Contribution:** 3 good
**Rating:** 5
**Confidence:** 3

**Summary:**

The paper tackles modeling bounded rational agents based on their trajectories. It focuses on determining $pi (a | s, R, \beta)$, where $R$ represents the reward function and $\beta$ denotes the computational budget. Contrasting the classical Boltzmann model that overlooks varying agent rationality, this work innovatively models $\beta$ as a variable derived from an agent-centric distribution $p_{\beta | \eta}$, with $\eta$ being inferred using MAP.

**Strengths:**

- **Originality**: One of the primary strengths of the paper is its original approach to model $\beta$ using a learnable parameter. This innovation distinguishes it from prior works, offering a fresh perspective in the domain of modeling bounded rational agents. The perspective is novel and fresh.

- **Significance**: The introduction of a learnable parameter for $\beta$ can potentially change the way we perceive agent behaviors. This approach, if refined and built upon, could pave the way for more advanced and adaptive models in the future. It also hints at the possibility of its application across different domains, making the work potentially impactful.

**Weaknesses:**

- **Clarity and Presentation**: The paper often comes across as convoluted, making it difficult for readers unfamiliar with the domain to grasp its core concepts. Clearer explanations, along with better-organized sections, would significantly improve comprehension.

- **Lack of Thorough Literature Review**: The paper does not delve deep into existing works, leaving readers unaware of the full landscape of related research. A dedicated literature review, even if placed in the appendix, would help contextualize the presented work better.

- **Quality of Experiments**: The experimental section lacks rigorous validation on diverse datasets or scenarios, or providing substantial comparisons with more state-of-the-art existing models or methodologies, potentially limiting the generalizability of the proposed method.

- **The Modeling of $\beta$**: The paper models $\beta$ in a manner that appears static across the trajectory. However, in realistic scenarios, it would be logical to assume that the distribution of $\beta$ is adaptive (either in a Bayesian or frequentist manner) and may change at every time step, conditional on histories. This richer representation can offer a more meaningful interpretation of agent behaviors.

- **Ambiguities in Parameter Inference**: While the idea of modeling $\beta$ with a learnable parameter is commendable, the paper does not sufficiently justify or explain the underlying mechanisms behind the chosen methodology for inferring $\eta$.

- **Robustness Concerns**: Given the intrinsic variability in agent behaviors, how robust is the proposed method to outliers or erratic trajectories? Were there specific tests or validations done in this regard? I suggest the authors to provide some ablation study in future revisions.

I am happy to raise my score if the authors address some of the concerns above.

**Questions:**

- **Literature Review**: Please provide a thorough literature review.

- **Future Directions**: For example, are there plans to address the adaptability of $\beta$ based on past trajectories? Or would you develop a rigorous theoretical guarantee for the model?

- **Practical Implications**: How does the authors envision the practical implications of this work? In which domains or applications do they see this method having the most significant impact? For example, one of the hottest research recently with real-world impact is on large language models. How may the classical bounded rational agents study make contributions to this direction?

- **Computational Complexity**: Can the authors comment on the computational complexity of their method, especially concerning the inference of $\eta$ (also please provide more details on the inference methodologies being used)? How scalable is it for large-scale applications?

---

> ### Author Response · Authors · 2023-11-17
> **Response to Reviewer q6an**
>
> Thank you for your review!
>
> **Clarity and Presentation:**
>
> We will try to improve clarity of presentation in our revision, specifically as it relates to the plots. Are there specific aspects that you would like to see improved?
>
> **Lack of Thorough Literature Review:**
>
> We discuss related work in section 2 as well as several others in sections 4-6. Are there papers that you think are missing or warrant further discussion?
>
> **Quality of Experiments:**
>
> We have included experiments in 3 very different domains as well as relevant state-of-the-art approaches for each of the domains. Are there specific comparisons with other models of bounded rationality that you think are missing?
>
> **Modeling of $\beta$:**
>
> This comment is not quite right: our approach does allow beta to vary per-step. However, we currently model these choices as independent; we do think that explicitly modeling state-dependence of beta is an interesting future direction.
>
> **Ambiguity in Parameter Inference:**
>
> $\eta$ represents the parameters of the model that produces the distribution over $\beta$. In all 3 domains, this denotes the parameters of a simple feedforward network which are learnt using gradient descent. We will make this clear in our revision.
>
> **Robustness Concerns:**
>
> We already include ablations in each of the three sections: Imitation Learning, Imitation Learning + Fixed Inference, Imitation Learning + Boltzmann Rational Inference and finally, Imitation Learning + Runtime Budget Inference  (which is our method). As evidenced by large differences in task success, there is a lot of variability in the real chess and language data, which in particular includes human players making weird or improbable moves in the lowest skill buckets. Nonetheless, L-IBMs are just as accurate in these buckets as the others.
>
> **Literature Review and future direction:**
>
> Discussed above.
>
> **Practical Implications:**
>
> We find this line of work to be an important contribution to the cognitive science and linguistics community as evidenced by the recent Nature paper [1]. Another key motivation for us was towards the development of better Human-AI systems. Like recent works like Diplomacy [2] points out, being able to model human behavior better would allow us to build AI systems (even, those that involve LLMs) that can better understand sub-optimal human behavior and therefore better cooperate with humans.
>
> [1] van Opheusden, Bas, et al. "Expertise increases planning depth in human gameplay." Nature (2023): 1-6.
>
> [2] Meta Fundamental AI Research Diplomacy Team (FAIR), et al. "Human-level play in the game of Diplomacy by combining language models with strategic reasoning." Science 378.6624 (2022): 1067-1074.
>
> **Computational Complexity:**
>
> Learning of $\eta$ is discussed above. We note that our method is more efficient than the alternatives as multiple values of $\beta$ can be evaluated as quickly as we can evaluate just one, because of the fact that we consider any-time algorithms. We also use stochastic gradient descent to learn $\eta$ and use inference algorithms like MCTS, RSA and BFS that are known to scale well in large-scale domains.

---

### Official Review · Reviewer_KCR8 · 2023-10-31

**Soundness:** 3 good
**Presentation:** 3 good
**Contribution:** 3 good
**Rating:** 6
**Confidence:** 4

**Summary:**

The paper presents budget constrained bounded rationality models. In contrast with classic Boltzman rationality, deviations from rationality are formalized as (unknown) limitations on computation instead of simply noise in selection. A simple formulation of budgets constructed as a model over possible inference budgets, and it is observed that inference can be computed efficiently for anytime models. The approach is demonstrated in three domains: inferring goals from actions in navigation, inferring communicative intent, and predicting chess moves. The budgets model individual performance and have meaningful interpretations.

**Strengths:**

- The paper presents an interesting and intuitive formalization of bounded rationality based on constrained computation.
- The empirical comparisons are nice.
- The results are interesting.

**Weaknesses:**

- There are a few ad hoc decisions buried in the middle, which make the story less clear.
- Comparisons with other proposals are a bit lacking. This is not the first paper to propose limitations on Boltzman rationality.
- A more detailed set of results would be nice.

Detailed comments:
- "consider again the trajectories depicted in Fig. 1(b–c), which differ only in the difficulty of the search problem, and not in the cost of the optimal trajectory at all." There are some pretty big assumptions hidden in here.
- "learning a model of these agents ultimately learning reward parameters θ and agent-specific budget-generating parameters" missing word?
- The sampling algorithm for the speech task doesn't seem to be motivated by the idea of a budget.
- I am not a fan of the visualization in figure 6. It is quite hard to parse.

**Questions:**

I would really love to see some additional comparisons. Other than that, I find the paper to be a nice contribution.

I would rate this a 7, but I don't seem to have that option...

---

> ### Author Response · Authors · 2023-11-17
> **Response to Reviewer KCR8**
>
> Thank you for your review!
>
> **Improving clarity:**
>
> Are there specific decisions that you would like us to elaborate on? We will work towards improving clarity of our presentation in our revision.
>
> **Comparison of other proposals:**
>
> Are there other alternative general proposals for these domains that you would like for us to compare to? In both Colors-in-context and Chess, we're comparing to what we believe to be the current state-of-the-art model. (In Chess, this is not just inferring a temperature parameter.)
>
> **Figure 1:**
>
> Fig. 1 was meant to be conceptual but we will improve the clarity of our assumptions in the next revision.
>
> **Typo:**
>
> Thank you for catching the typo. We have fixed this in our revision.
>
> **RSA:**
>
> In colors-in-context, we consider the case where runtime is truncated at a fixed # of iterations. You can do more RSA iterations if you want, but on this task as past work [1, 2] has shown it doesn't make a difference.
>
> [1] McDowell, Bill, and Noah Goodman. "Learning from omission." Proceedings of the 57th Annual Meeting of the Association for Computational Linguistics. 2019.
>
> [2] Monroe, Will, et al. "Colors in context: A pragmatic neural model for grounded language understanding." Transactions of the Association for Computational Linguistics 5 (2017): 325-338.
>
> **Figure 6:**
>
> We have improved Figure 6 in our current revision. Please let us know if it is better.

---

> > ### Comment · Reviewer_KCR8 · 2023-11-22
> > **Follow up**
> >
> > Thanks for your response. Here is one example of modeling human bounded reasoning:
> > https://dl.acm.org/doi/abs/10.1145/3319502.3374832
> > I believe Dorsa Sadigh has other work along these lines as well. The paper is not about chess, but I took your goal to be more general than playing chess?
> >
> > For the strong assumptions point: you have italicized "only". Since we are interested in human limitations, that "only" involves making some idealizations that may not make sense.
> >
> > Regarding RSA, I'm not sure what you mean "doesn't make a difference". It does matter, there have been theoretical analyses. Perhaps you could be a bit more precise in what sense it does not matter for your interests?

---

> > > ### Author Response · Authors · 2023-11-22
> > > **Follow up to Reviewer KCR8**
> > >
> > > Thank you for your reply! We really appreciate the additional constructive questions, allowing us to strengthen the paper before the final revision.
> > >
> > > 1) Yes, there are definitely methods that have been developed for specific applications. The paper you cited is interesting and relevant, but not directly applicable to the tasks we're studying. It assumes that rewards have fixed, meaningful scales known and communicated to human players, then studies deviation from optimal policy under those rewards. In our work, because we are using a 0-1 reward (in RSA) and a learned reward model (in chess), it will not make any predictions that are different from the base Boltzmann model. But, we do think extending our approach to accommodate CPT-type models would be an interesting direction for future work! We will add a discussion relating to this work in our final revision.
> > >
> > > 2) Thanks for clarifying! We emphasize that this is a conceptual figure, and that the specific modeling assumptions made by our approach are described in the relevant experiment sections. Like all models, our L-IBMs simplify aspects of the human decision-making process; their validity is ultimately validated by their empirical success in predicting human behavior and inferring meaningful latent skill parameters. We will make this clearer in our final revision.
> > >
> > > 3) Our earlier response wasn't clear. It is indeed true that RSA iterations do make a difference in general on standard implicature tasks. However, in colors-in-context, the two prior studies listed above have shown that increasing depth does not increase **human predictivity** on these tasks.
> > >
> > > We hope that we have addressed your questions.

---

### Official Review · Reviewer_egSD · 2023-10-31

**Soundness:** 4 excellent
**Presentation:** 4 excellent
**Contribution:** 3 good
**Rating:** 8
**Confidence:** 4

**Summary:**

In this work, the authors address the problem of modeling agents with bounded rationality and possibly unknown preferences. They begin by pointing out that the standard Boltzmann model depends only on the return of a particular action and doesn’t account for the structure of a problem in determining the probability of a suboptimal outcome given a particular inference budget for an agent.

In (2), the agents present L-IBM, a MAP inference algorithm that allows us to infer the reward model and the parameters of the budget distribution for a particular agent from data assuming the agent follows an anytime inference algorithm. Then in the following sections they give instantiations of their setup in maze solving, language generation, and chess.

In maze solving, they assume the search algorithm is a truncated BFS algorithm with a known heuristic. They show that this is an anytime inference algorithm and then demonstrate on simulated inference data that the L-IBM method is able to entirely recover the correct data-generating parameters while the Boltzman model performs horribly.

In language, they use a reference game–where one participant must try to communicate with another effectively and must model their understanding well enough to communicate. Here, they borrow a model of Bayesian listeners and speakers from the cognitive science literature. The primary feature of interest in this model is the number of “layers deep” to go in modeling that your partner can model that you know things and you can model that your partner can model that you know things and so on. They again show that this is an anytime inference model and demonstrate using a transformer and an existing dataset of human utterances and choices. Here, the bounded rationality model is able to determine that more skilled players behave in ways that would be considered “deeper” than those that are less skilled.

Here, the anytime inference algorithm is MCTS as used in AlphaGo and the recent Diplomacy works. In this setting, there are two budget parameters \beta_UCT, and \beta_runtime. The method is used to estimate each of these parameters. Using a dataset of human games of players with varying Elo ratings and time controls, the quality of moves is inferred to these two beta parameters and they are shown to correlate with a longer time control, stronger opponent, and stronger player (as one would expect a priori).

**Strengths:**

A substantive assessment of the strengths of the paper, touching on each of the following dimensions: originality, quality, clarity, and significance. We encourage reviewers to be broad in their definitions of originality and significance. For example, originality may arise from a new definition or problem formulation, creative combinations of existing ideas, application to a new domain, or removing limitations from prior results. You can incorporate Markdown and Latex into your review. See https://openreview.net/faq.

I think this paper tackles an interesting question and gives a thorough and principled answer. As we introduce other cognitive agents into the world we ought to have a framework for evaluating their and our behavior on the same terms. This feels like a step towards a more general understanding of this. I think the boundedly rational agents as described in this paper are a substantial generalization of the Boltzmann model.

The evaluation on 3 domains is thorough and I appreciated the care taken to make them easy to understand and well presented. I found the figures and data presentation to be compelling and the ideas presented have had me update my model of how to think about this kind of thing going forward.

**Weaknesses:**

A substantive assessment of the weaknesses of the paper. Focus on constructive and actionable insights on how the work could improve towards its stated goals. Be specific, avoid generic remarks. For example, if you believe the contribution lacks novelty, provide references and an explanation as evidence; if you believe experiments are insufficient, explain why and exactly what is missing, etc.

At some point in the beginning of the paper the problem statement seemed too general to grab on to and I was mentally scraping around for how to add structure in order to come up with ideas for solutions. I think it would be a nicer reading experience to include a comment gesturing in the direction of anytime inference algorithms in the front part of the paper so as to anticipate this.


I’m not sure how to interpret the numerical figures for something like chess in Figure 5. Naively, I tend to think that the effects on predicting next actions are not very large.

It is not clear to me how much this can be used in the broad main thrusts of ML research today and this work might be of limited use to practitioners. However, I think it is interesting and probably valuable for subjects like reward-free Offline RL.

**Questions:**

How hard is it in general to predict the player’s next move in chess?
Is there an extension of anytime inference that can handle “amortized computations” like dynamic programming for value functions or other pre-planning methods?
What did you practically use to implement the inference procedures in this paper? I think the methods would be useful to comment on a bit.

---

> ### Author Response · Authors · 2023-11-17
> **Response to Reviewer egSD**
>
> Thank you for your review!
>
> **Introducing anytime inference algorithms earlier:**
>
> Thank you for the suggestion! We will introduce anytime algorithms earlier in the paper to make it easier to follow as well as work towards improving the clarity of our presentation in our next revision.
>
> **Magnitude of improvement:**
>
> 1. Chess being a sequential game, small improvements in per-step prediction accuracies can often be significant over the course of the game due to cascading errors.
> 2. In scientific applications, we often care about inferring skill parameters for their own sake, and not just to improve predictive accuracy. In this sense, we view it as a strength of L-IBM that it can infer these population-level parameter differences from only small differences in behavior. Indeed, in both language understanding & chess, we're not aware of past work that can automatically derive these differences from behavior alone; the ability to do so has direct applications in cognitive science [1] and computational linguistics [2].
>
> [1] van Opheusden, Bas, et al. "Expertise increases planning depth in human gameplay." Nature (2023): 1-6.
>
> [2] Franke, Michael, and Judith Degen. "Reasoning in reference games: Individual-vs. population-level probabilistic modeling." PloS one 11.5 (2016): e0154854.
>
> **Applications:**
>
> In addition to the relevance to the RL community that you pointed out, we also find this line of work to be important to the cognitive science and linguistics community as evidenced by a recent Nature paper [1] and others [2]. As it relates to the ML community, one of our key motivations was towards the development of better Human-AI systems. Like recent works like Diplomacy [3] points out, being able to model human behaviour better would allow us to build AI systems that can better understand sub-optimal human behaviour and therefore better cooperate with humans.
>
> [1] van Opheusden, Bas, et al. "Expertise increases planning depth in human gameplay." Nature (2023): 1-6.
>
> [2] Jeon, Hong Jun, Smitha Milli, and Anca Dragan. "Reward-rational (implicit) choice: A unifying formalism for reward learning." Advances in Neural Information Processing Systems 33 (2020): 4415-4426.
>
> [3] Meta Fundamental AI Research Diplomacy Team (FAIR), et al. "Human-level play in the game of Diplomacy by combining language models with strategic reasoning." Science 378.6624 (2022): 1067-1074.
>
>
> **How hard is it in general to predict the player’s next move in chess?:**
>
> The task of predicting the player's next move in chess is still a challenging and open problem given the size of the action spaces as well as other components that influence them like player styles and partner styles. Note, that even state of the art models in chess [1, 2] are still far from perfect accuracy.
>
> [1] McIlroy-Young, Reid, et al. "Aligning superhuman ai with human behavior: Chess as a model system." Proceedings of the 26th ACM SIGKDD International Conference on Knowledge Discovery & Data Mining. 2020.
>
> [2] Jacob, Athul Paul, et al. "Modeling strong and human-like gameplay with KL-regularized search." International Conference on Machine Learning. PMLR, 2022.
>
> **Is there an extension of anytime inference that can handle “amortized computations” like dynamic programming for value functions or other pre-planning methods?**
>
> Yes, and we're doing it already in the first experiment. We model maze exploration using BFS; BFS is just a special case of Dijkstra's algorithm, which is a classic dynamic programming algorithm. We can think of the inference states $f_i$ as holding the state of a dynamic program. In fact, value iteration is an anytime algorithm as well, where the inference states $f_i$ holds the current values for all states.
>
> **What did you practically use to implement the inference procedures in this paper? I think the methods would be useful to comment on a bit.**
>
> ##### Maze Domain:
> We implemented truncated BFS using Pytorch and Numpy and we also used the mazelib library to generate the data.
>
> ##### Colors-in-context Domain:
> We trained the models using Pytorch and Huggingface libraries. We specifically implemented RSA using Pytorch.
>
> ##### Chess Domain:
> For chess, the base policy and value functions were trained using Ray library and Pytorch. MCTS was specifically implemented using Numpy. We also used the pettingzoo library for simulating moves.
>
> We have added these details to our revision and also plan to release all of our code publicly.

---

### Official Review · Reviewer_PxXG · 2023-11-01

**Soundness:** 3 good
**Presentation:** 3 good
**Contribution:** 2 fair
**Rating:** 6
**Confidence:** 3

**Summary:**

The paper proposes a method for modeling bounded rational agents through the inference of latent budget. The method seeks to model an agent's computationally constrained inference by explicitly inferring the latent variable associated with the computational budget jointly with the agent's goals. This is used to both infer agent intent as well the underlying budget, which is correlated with agent competency. The proposed method is experimentally evaluated over three tasks: maze navigation, language understanding, and chess.

**Strengths:**

* The paper is well-written and motivated. The variety of experimental settings is helpful for gauging model performance, and in particular the inclusion of an experiment with human-generated data (RSA) is valuable in showing performance when the exact underlying agent model is unknown.
* The inference of agent budget and the correlation to skill is an interesting direction, and may have value in both agent-agent and human-agent interactions.

**Weaknesses:**

* The accuracy differences in the RSA and chess tasks are fairly marginal, and seem to indicate that it is difficult to jointly infer both latent budgets and intent in more complicated tasks.
* The modeling of the latent budget requires fairly strong assumptions about the underlying reasoning mechanism of the agent. In addition to potential misalignment of assumptions, this also leads to situations like Sec. 5.3 where a constant of proprotionality is approximated over the set of all natural strings which are potential additional sources of error.
* The results graphs are difficult to parse with the frequent reference to variable names that have unintuitive meanings. I realize these are referring to parameters in the agent models and corresponding sub-populations, but it is difficult for the reader to draw conclusions when the results are represented in terms of abstract terms such as beta_temp, beta_depth, beta_runtime, beta_puct, etc. This is compounded by the fact that terms like "depth" are used frequently and seem to have context-specific meanings that are not always fully defined.

**Questions:**

1) What exactly does beta_temp intuitively represent in the RSA task? This is not entirely clear to me, nor why the figures in Table 3a/b seem to show such uninformative results. Do you have any insight as to why the inferred beta_depth seems to be more informative than beta_temp (in the sense that it seems more correlated with player skill)?
2) What is the relationship between beta_puct and ELO rating in Sec. 6? For time control it seems reasonable that longer time budgets would enable more exploration, but it seems there would be a trade-off between exploration and exploitation with respect to ELO rating. So it's not clear to me what this relationship is expected to be.

---

> ### Author Response · Authors · 2023-11-17
> **Response to Reviewer PxXG**
>
> Thanks for the review!
>
> **Accuracy differences:**
>
> 1. Although the performance improvements are small in an absolute sense, in chess (which is sequential in nature), these improvements at a per-move level can quickly cascade at a trajectory level.
> 2. In scientific applications, we often care about inferring skill parameters for their own sake, and not just to improve predictive accuracy. In this sense, we view it as a strength of L-IBM that it can infer these population-level parameter differences from only small differences in behavior. Indeed, in both language understanding & chess, we're not aware of past work that can automatically derive these differences from behavior alone; the ability to do so has direct applications in cognitive science (e.g., [1]) and computational linguistics (e.g., [2]).
>
> [1] van Opheusden, Bas, et al. "Expertise increases planning depth in human gameplay." Nature (2023): 1-6.
>
> [2] Franke, Michael, and Judith Degen. "Reasoning in reference games: Individual-vs. population-level probabilistic modeling." PloS one 11.5 (2016): e0154854.
>
> **Modeling Assumption:**
>
> 1. The key underlying assumption is that the inference procedure is an anytime algorithm. While not allowing arbitrary inference procedures, this still accommodates a wide variety of algorithms that are used in practice. Most importantly, even with these assumptions, our method successfully explains real behavioral data from humans.
> 3. The challenge in computing the normalizing constant in Sec. 5.3 is a property of all rational speech acts models, and not inherent to to latent budget modeling. As reference games admit an arbitrarily large natural language utterance set, it is indeed the case that we have to include some form of approximation. We use the same approximation that past work has shown to be effective. [1, 2]
>
> [1] McDowell, Bill, and Noah Goodman. "Learning from omission." Proceedings of the 57th Annual Meeting of the Association for Computational Linguistics. 2019.
>
> [2] Monroe, Will, et al. "Colors in context: A pragmatic neural model for grounded language understanding." Transactions of the Association for Computational Linguistics 5 (2017): 325-338.
>
> **Improving Clarity of Plots**
>
> In our uploaded revision, we have improved the clarity of the plots to make them easier to parse. Please let us know if this is better!
>
> **$\beta_\mathrm{temp}$ and $\beta_\mathrm{runtime}$ in RSA:**
>
> Intuitively, $\beta_\mathrm{temp}$ can be seen as a greediness parameter. A low $\beta_\mathrm{temp}$ indicates that the speaker is optimally "best-responding" to the listener and a high $\beta_\mathrm{temp}$ indicates that the speaker is choosing actions from a uniform distribution. Empirically, human language users are somewhere between the two extremes, and past work has found that tuning $\beta_\mathrm{temp}$ is necessary to get a good fit to human data [1]. Despite this, we find here that *individual human differences* in language production are not explained by *differences in $\beta_\mathrm{temp}$*, but they are explained by an L-IBM (inferring $\beta_\mathrm{runtime}$) that models individual participants as literal or pragmatic listeners.
>
> [1] Frank, Michael C., and Noah D. Goodman. "Predicting pragmatic reasoning in language games." Science 336.6084 (2012): 998-998.
>
>
> **Relationship between $\beta_\mathrm{puct}$ and ELO rating:**
>
> $\beta_\mathrm{puct}$ (as used in popular strength-modeling approaches like [1]) doesn’t simply control exploration vs. exploitation. Instead, it biases exploration towards a specific policy prior. As $\beta_\mathrm{puct}$ tends to infinity, it is identical to playing the prior policy (effectively reducing to imitation learning). When $\beta_\mathrm{puct}$ is set to 0, it is equivalent to greedily picking the search Q-values. As it relates to ELO rating, what we are noticing here is that stronger players start deviating more from their base policy to instead depend more on their MCTS search Q-values. This is therefore indicating that stronger players rely *more* on search compared to weaker players.
>
> We have added this discussions to our revision!
>
> [1] Jacob, Athul Paul, et al. "Modeling strong and human-like gameplay with KL-regularized search." International Conference on Machine Learning. PMLR, 2022.

---

> > ### Comment · Reviewer_PxXG · 2023-11-21
> >
> > Thanks for the detailed reply.
> >
> > The additional clarification regarding the relationships between the beta parameters is appreciated. Although I'm not sure the clarity of the plots has been improved too much (the titles help, but doesn't address the issue that these names are very unintuitive to begin with).
> >
> > Regarding your chess comment on sequential improvement, this an interesting point. Is there anything to suggest that this performance improvement is actually significant at that level and cascades as you suggest, or is this conjecture?

---

> > > ### Author Response · Authors · 2023-11-22
> > > **Response to Reviewer PxXG**
> > >
> > > Thank you so much for your response! We acknowledge that the various parameters and the corresponding variables could be unintuitive. In our final revision, we will include a table defining all the rationality parameters that we consider in our work for each algorithm. Would this be helpful?

---

### Author Response · Authors · 2023-11-17
**Note from Authors**

Thank you all for your thorough reviews!

We’re glad that you found the paper “well-written and motivated” (PxXG), “thorough and principled” (egSD), “interesting and intuitive” (KCR8), and “original” (q6an). We believe that your suggestions have helped improve the manuscript. In our new revision, we have improved our figures and tables.

In the next few days, we will submit another revised version that incorporates your comments.

---

### Author Response · Authors · 2023-11-20
**Note from Authors**

Dear Reviewers,

As the discussion period is about to close, please let us know if there is anything else you would like to see.
Thank you!

---

### Public Comment · ~Tan_Zhi-Xuan1 · 2024-04-04
**Relevant Prior Work**

Belatedly discovering this (very neat) paper! It's great to see more people working on boundedly rational agent models.

I wanted to point the authors towards some highly relevant prior work that they may have missed, and that they may want to cite in future revisions or extensions of this paper:

1. [Learning the preferences of bounded agents.](https://owainevans.github.io/pdfs/preferences_bounded_agents_evans.pdf) (Evans & Goodman, 2015)
    - In this workshop paper, Evans & Goodman (2015) infer the utility functions under the assumption of myopic/depth-limited planning, among other computational bounds. While their experiments are not as extensive, one interesting difference from L-IBM is that they *jointly* infer the budget/bound parameter alongside the Boltzmann temperature parameter. The planning algorithm they use can be viewed as form of truncated breadth-first search over the AND-OR graph of the MDP, with the optional use of Monte Carlo samples to handle transition uncertainty (in which case the algorithm becomes more like MCTS).
2. [On the Feasibility of Learning, Rather than Assuming, Human Biases for Reward Inference](https://proceedings.mlr.press/v97/shah19a.html) (Shah et al, 2019)
    - This paper considers how to learn the reward functions of agents with unknown cognitive biases. Unlike L-IBM, they do not assume a specific algorithm or latent computational bound to perform Bayesian inference over, but instead train a value iteration network via standard maximum likelihood training to try and automatically capture various biases. They also test this algorithm against data generated by a myopic planner.
3. [Online Bayesian Goal Inference for Boundedly Rational Planning Agents](https://proceedings.neurips.cc/paper/2020/hash/df3aebc649f9e3b674eeb790a4da224e-Abstract.html) (Zhi-Xuan et al, 2020)
    - In this paper, we introduced a class of boundedly rational agent models similar to the truncated BFS and MCTS algorithms in your paper, for the purposes of goal inference from suboptimal plans (similar to your task in Section 4, and the example given in Figure 1). Similar to L-IBM, we place a prior over the search budget of a best-first / A* search algorithm, and use that to simulate a whole range of possible plans that the agent might execute. Unlike L-IBM, we assume that this budget is independently sampled each the time the agent forms a new plan, rather than being a "static" parameter that is sampled at the start. This means that observers cannot make inferences like "this agent tends to plan myopically because I keep seeing it make mistakes", since our model assumes that the search budget keeps being resampled.
   - Of course, there are hybrids between L-IBM and our model one could try, like placing a hyper-prior over the average search budget, while still having the agent sample random search budgets when it starts planning.
4. [Modeling the Mistakes of Boundedly Rational Agents Within a Bayesian Theory of Mind](https://arxiv.org/abs/2106.13249) (Alanqary et al, 2021)
   - In this follow up paper to the above, we more systematically tested how our boundedly rational agent model can account for various kinds of mistakes (planning mistakes vs. action mistakes, etc.) that an agent might make, showing how the use of boundedly rational model leads to more human-like inferences, in contrast to the Boltzmann rational model (which only models action mistakes).

[Separately, I noticed that the title of Section 4 currently seems to conflict with the text. Is the algorithm "Truncated Depth-First Search" (what the section title says), or "Truncated Breadth-First Search" (what the text says)?]

If the authors are interested in working on extensions of this paper, feel free to reach out . We have a bunch of infrastructure that would make it relatively straightforward to scale L-IBM in various directions, including the hierarchical Bayesian extensions mentioned above, but also coverage of a wider range of planning algorithms (A*, RTDP / Real-Time Heuristic Search, MCTS, etc.).

---

### Meta-Review · Area_Chair_odX5 · 2023-12-10

**Metareview:**

a) Claims: The paper introduces the idea of modeling bounded rationality by a bound on the runtime of an inference algorithm, rather than Boltzmann noise.  An algorithm for estimating the budget is presented and evaluated on three different domains.

b) Strengths: The reviewers agreed that this is an interesting approach to formalizing "rationality" that has advantages over the Boltzmann noise model.  Many reviewers found the choice and variety of evaluation domains valuable.

c) Weaknesses: The main concerns raised by reviewers were whether the improvements in prediction accuracy were significant (in the non-statistical sense), and whether the paper sufficiently engaged with existing work in the literature.  All reviewers also found the figures difficult to parse.

**Justification For Why Not Higher Score:**

My main motivation for not recommending a higher score is the concerns about the usefulness/significance of the results.

**Justification For Why Not Lower Score:**

The paper presents an interesting approach to the important and tricky issue of formally modeling bounded cognition and/or rationality.

---

### Decision · Program_Chairs · 2024-01-16

Accept (poster)